# Screening and Identification of Lassa Virus Entry Inhibitors from a Fragment-Based Drug Discovery Library

**DOI:** 10.3390/v14122649

**Published:** 2022-11-27

**Authors:** Yuxia Hou, Yang Liu, Xiaoying Jia, Minmin Zhou, Wenting Mao, Siqi Dong, Yueli Zhang, Gengfu Xiao, Wei Wang

**Affiliations:** 1State Key Laboratory of Virology, Center for Biosafety Mega-Science, Wuhan Institute of Virology, Chinese Academy of Sciences, Wuhan 430071, China; 2University of the Chinese Academy of Sciences, Beijing 100049, China; 3State Key Laboratory of Medicinal Chemical Biology, College of Pharmacy, Nankai University, Tianjin 300071, China

**Keywords:** Lassa virus (LASV), fragment-based drug discovery (FBDD) library, glycoprotein complex (GPC), membrane fusion, transmembrane domain (TM)

## Abstract

Lassa virus (LASV) is a highly pathogenic virus that is categorized as a biosafety level-4 pathogen. Currently, there are no approved drugs or vaccines specific to LASV. In this study, high-throughput screening of a fragment-based drug discovery library was performed against LASV entry using a pseudotype virus bearing the LASV envelope glycoprotein complex (GPC). Two compounds, F1920 and F1965, were identified as LASV entry inhibitors that block GPC-mediated membrane fusion. Analysis of adaptive mutants demonstrated that the transient mutants L442F and I445S, as well as the constant mutant F446L, were located on the same side on the transmembrane domain of the subunit GP2 of GPC, and all the mutants conferred resistance to both F1920 and F1965. Furthermore, F1920 antiviral activity extended to other highly pathogenic mammarenaviruses, whereas F1965 was LASV-specific. Our study showed that both F1920 and F1965 provide a potential backbone for the development of lead drugs for preventing LASV infection.

## 1. Introduction

Lassa virus (LASV) belongs to the genus *Mammarenavirus* (family *Arenaviridae*) [1]; it causes the hemorrhagic disease Lassa fever (LF), an annual epidemic in West Africa that peaks during the dry season. In 2021, Nigeria faced a large outbreak of LF, with 510 confirmed cases and 102 deaths, and a case fatality ratio (CFR) of 20.0%, according to the Nigeria Centre for Disease Control. LASV, as most pathogenic mammarenaviruses, including the Junín virus (JUNV), Machupo virus (MACV), Guanarito virus (GTOV), Chapare virus (CHAPV), Sabiá virus (SBAV), and Lujo virus (LUJV), cause severe hemorrhagic fever and are listed as biosafety level-4 (BSL-4) agents.

LASV is an enveloped, bi-segmented RNA virus with an S genome that encodes the glycoprotein complex (GPC) and nucleoprotein, and an L genome that encodes the matrix protein (Z) and the viral polymerase. GPC contains three subunits: the stable signal peptide (SSP), receptor-binding subunit GP1, and membrane fusion subunit GP2. Most recently, the complete native LASV GPC spike was deposited, and it was found that SSP crosses the membrane once and helps stabilize GPC in its native conformation [2]. In the membrane-proximal and transmembrane domains, SSP interacts with GP2 and forms a platform targeted by most fusion inhibitors [3,4,5,6].

To date, no drug- or vaccine-specific LASV has been developed. Therapeutic strategies are limited to ribavirin administration in the early stages of illness. Notably, LHF-535, an entry inhibitor that targets the arenavirus GPC and inhibits the membrane fusion, has completed phase 1 clinical studies and is in development as a therapeutic candidate to treat LF and other hemorrhagic fevers of arenavirus origin [7,8]. We have screened multiple drug libraries to identify an effective fusion inhibitor targeting arenavirus GPC [4,9,10,11,12]. In this work, we screened a fragment-based drug discovery (FBDD) library to identify novel backbone compounds. Using a pseudotype of LASV in BSL-2 facilities, we determined that two compounds, F1920 and F1965, inhibited LASV entry.

## 2. Materials and Methods

### 2.1. Cells Lines

BHK-21, BSR-T7, HEK 293T, Vero, and A549 cells were cultured in Dulbecco’s modified Eagle’s medium (DMEM; HyClone, Logan, UT, USA) supplemented with 10% fetal bovine serum (Gibco, Grand Island, NY, USA) at 37 °C with 5% CO_2_.

### 2.2. Viruses

The pseudotype VSV bearing the GPC of *Mammarenaviruses* was generated as previously reported [11]. Further, the LASV GPC genes from the Josiah strain (GenBank HQ688673.1); lymphocytic choriomeningitis virus (LCMV) GPC genes from Armstrong strain (GeneBank AY847350.1); LUJV CPC genes (GenBank NC_012776.1); Mopeia Virus (MOPV) GPC genes (GenBank AY772170.1); GTOV GPC genes (GenBank NC_005077.1); JUNV GPC genes from XJ13 strain (GeneBank NC_005081.1); MACV GPC genes from Carvallo strain (GeneBank NC_005078.1); SBAV GPC genes (GeneBank U41071.1), and CHAPV GPC genes (GenBank NC_010562.1) were cloned into the mammalian expression vector pCAGGS. Similarly, including Ebola virus (EBOV) GPC genes (strain Mayinga, EU224440.2) and Marburg virus (MARV) GPC genes (GenBank YP_001531156.1) of *Filoviridae*, their GPCs were also cloned into pCAGGS vector. Then, 293T cells transfected with pCAGGS GPC were infected at an MOI of 0.1 for 1 h with pseudotype VSV (described below) in which the G gene was replaced with a luciferase gene. The culture supernatants were harvested 24 h later, centrifuged to remove cell debris, and stored at −80 °C.

The recombinant VSV expressing LASV GPC was generated as described previously [11,13].The plasmid used for constructing the recombinant virus was pVSVΔG-eGFP (where eGFP is enhanced green fluorescent protein) (plasmid 31842; Addgene, Watertown, MA, USA). GPC was cloned into the ΔG site, and the construct was designated pVSVΔG-eGFP-GPC. BHK-21 cells grown in 6-well plates were infected with a recombinant vaccinia virus (vTF7-3) encoding T7 RNA polymerase at an MOI of 5. After 45 min, cells were transfected with 11 μg of mixed plasmids with a 5:3:5:8:1 ratio of pVSVΔG-eGFP-GPC (pVSVΔG-Rluc for generating pseudotype VSV), pBS-N, pBS-P, pBS-G, and pBS-L, respectively.

The pseudotype and recombinant viruses bearing LASV GPC with VSV backbone were designated LASV_VSV_pv and LASV_VSV_rv, respectively. The titer of LASV_VSV_pv was measured by infecting BHK-21 cells previously transfected with pCAGGS-VSV G and determined by plaque assay 24 h post-infection. The titer of LASV_VSV_rv was determined by plaque assay. The titers of LASV_VSV_pv and LASV_VSV_rv were 2 × 10^7^ PFU/mL and 3 × 10^7^ PFU/mL, respectively.

Plasmid-based virus rescue of LCMV was carried out as previously reported [14,15]. LCMV clone 13 genome RNA L (GeneBank DQ361066) and S (Genebank DQ361065) segments were used to construct the plasmids pT7-LCMV-L and pT7-LCMV-S, respectively. BSR-T7 cells were transfected with the mixed plasmids of pCAGGS-LCMV-NP (300 ng), pCAGGS-LCMV-L (600 ng), pT7-LCMV-L (600 ng), and pT7-LCMV-S (300 ng). The supernatant was collected 72 h later and inoculated into BHK-21 cells for amplification. The titer of LCMV Cl13 was 2 × 10^5^ PFU/mL by plaque assay.

The recombinant LCMV expressing LASV GPC was generated. BSR-T7 cells were grown in 6-well plates and were transfected with the mixed plasmids of pCAGGS-LCMV-NP (2.5 μg), pCAGGS-LCMV-LP (0.5 μg), pCAGGS-LASV GPC (0.4 μg), pT7-LCMV-L (3.0 μg), and pT7-rLCMV (GFP-P2A-NP)-LASV GPC (1.5 μg) by using 5 μL Lipofectamine 3000 for 4 h. After 4 days, these cells were seeded in T25 for extended culture, and the green fluorescence was observed on the eighth day, the supernatant was collected on ninth day, then inoculated into BHK-21 cells for amplification for about 6–7 days. The titer of LASV_LCMV_rv was 1 × 10^5^ PFU/mL by TCID_50_ assay.

### 2.3. Optimization of High-Throughput Screening (HTS) Assay Conditions

Cell density was optimized at 2 × 10^4^/well, and Vero cells were infected with MOI of 0.05 per well in 96-well plates. Methyl-beta-cyclodextrin (MβCD; 2 mM) and 0.5% DMSO were used as the positive and negative controls, respectively. The signal-to-basal (S/B) ratio and Z′ factor were determined as 1709 and 0.65, respectively.

### 2.4. HTS Assay of the Fragment Drug Library

A library of 1015 fragment-based drugs was purchased from Selleck Chemicals (Cat: L1600; Houston, TX, USA). Compounds were stored in 10 mM stock solution in DMSO at −80 °C until use. HTS was carried out as shown in Figure 1A. Vero cells were treated in duplicate with the compounds (100 μM); 1 h later, cells were infected with LASV_VSV_pv (MOI, 0.05) for 1 h. After 23 h, the infected cells were lysed, and luciferase activity was measured using the Renilla luciferase (Rluc) assay system (Promega, Madison, WI, USA). Cell viability was evaluated by using the Cell Counting Kit-8 (CCK-8) assay. In order to determine the IC_50_, LASV_VSV_pv at an MOI of 0.05 was used utilizing the timeline described above. Luciferase activity was measured using the Rluc assay system, and IC_50_ was calculated using GraphPad Prism 8.

### 2.5. Membrane Fusion Assay

293T cells co-transfected with pCAGGS-LASV GPC and pEGFP-N1 were treated with either compounds or vehicle (0.4% DMSO) for 2 h, followed by incubation for 15 min with acidified (pH 5.0) medium. The cells were then placed in neutral medium, and syncytium formation was observed by fluorescence microscope after 2 h.

For quantification of the luciferase-based fusion assay, 293T cells in 24-well plates transfected with both pCAGGS-LASV GPC (0.25 μg) and plasmids expressing T7 RNA polymerase (pCAGT7, 0.25 μg) were cocultured at a ratio of 3:1 with targeted cells transfected with pT7EMCVLuc (2.5 μg per well in 6-well plates) and 0.1 μg pRL-CMV (plasmids used in this assay were kindly provided by Yoshiharu Matsuura, Osaka University, Osaka, Japan). After 6 h of incubation, the compound treatment and pH induction were conducted as described above. Cell fusion activity was quantitatively determined after 4 h by measuring firefly luciferase activity expressed by pT7EMCVLuc and was standardized with Rluc activity expressed by pRL-CMV by using the Dual-Glo luciferase assay (Promega, Madison, WI, USA).

### 2.6. Virucidal Assay

To study virucidal effects, approximately LASV_VSV_pv or VSVpv in an MOI of 20 was incubated with hit compounds (400 μM) or vehicle at 37 °C for 1 h. The mixture was diluted 400-fold to a non-inhibitory concentration (1 μM) to infect Vero cells in a 96-cell plate. At the same time, Vero cells were treated in duplicate with the hits (1 μM); 1 h later, cells were infected with LASV_VSV_pv or VSVpv (MOI, 0.05) for 1 h, luciferase activity was determined 23 h later, as described above.

### 2.7. IIH6 Inhibition Assay

A549 cells were pretreated with either F1920 or F1965 of indicated concentrations (50 μM, 100 μM, 200 μM) at 37 °C for 1 h. As a control, A549 cells were pretreated either with 200 μg/mL IIH6 (sc-53987; Santa Cruz Biotechnology, Dallas, TX, USA) or a control IgM (50 μg/mL) at 4 °C for 1 h. The cells were then transferred to ice, and LASV_VSV_rv (MOI, 0.5) was added for 1 h. After three washes with cold phosphate-buffered saline (PBS), the cells were incubated at 37 °C for 12 h. GFP-positive cells were counted using an Operetta high-content imaging system. Nine fields per well were imaged on an Operetta high-content imaging system (PerkinElmer, Waltham, MA, USA), and the percentages of infected and DAPI-positive cells were calculated using Harmony 3.5 (PerkinElmer, Waltham, MA, USA).

### 2.8. Selection of Adaptive Mutants

Drug-resistant viruses were generated by passaging LASV_VSV_rv in Vero cells in the presence of 200 μM F1920 or F1965. LASV_VSV_rv was also passaged in the presence of 0.2% DMSO in parallel as a control. RNA from the resistant viruses was extracted using TRIzol (TaKaRa Bio, Inc, Kusatsu, Japan) and reverse transcribed using the PrimeScript RT reagent kit. The GPC segment was amplified and sequenced as previously described. Mutant sites were introduced to LASV_VSV_pv as previously described, and then F1920 and F1965 sensitivities were determined by Rluc activity.

### 2.9. Inhibition of F1920 or F1965 against Authentic LCMV Infection

Vero cells were seeded at 1 × 10^5^ cells per well in a 24-well plate, and after incubating overnight, Vero cells were incubated in the absence and presence of F1920 or F1965. After 1 h, the LCMV Cl13 was added at an MOI of 0.01. Then, the supernatants were replaced with the hit compounds for 23 h. Cell lysates were subjected to real-time (RT) quantitative PCR (qPCR) using the primers. The primer sequences used were as follows: LCMV-F: 5′-AGAATCCAGGTGGTTATTGCC-3′, LCMV-R: 5′-GTTGTAGTCAATTAGTC GCAGC-3′, GAPDH-F: 5′-TCCTTGGAGGCCATGTGGGCCAT-3′, and GAPDH-R: 5′-TGATGACATCAAGAAGGTGGTGAAG-3′. The RT-qPCR procedure was set up as follows: 50 °C for 15 min, 85 °C for 5 s, 95 °C for 1 min followed by 40 cycles consisting of 95 °C for 10 s, 60 °C for 20 s.

### 2.10. Inhibition of F1920 or F1965 against Recombinant Viruses Infection

Vero cells were seeded at 2 × 10^4^ cells per well in a 96-well plate, after incubating overnight, Vero cells were incubated in different concentrations of F1920 or F1965 for 1 h, and then LASV_VSV_rv was added at an MOI of 0.05 for additional 1 h, the supernatants were then replaced with either hit for 23 h. Similarly, Vero cells were incubated with F1920 or F1965 for 1 h, and then LASV_LCMV_rv was added at an MOI of 0.05 for additional 2 h. The supernatants were replaced with either hit for 23 h. Then, the GFP-positive cells were counted using an Operetta high-content imaging system. Nine fields per well were imaged on an Operetta high-content imaging system (PerkinElmer, Waltham, MA, USA), and the percentages of infected and DAPI-positive cells were calculated using Harmony 3.5 (PerkinElmer).

## 3. Results

### 3.1. Screening of an FBDD Library for Entry Inhibitors against LASV

To study viral entry in BSL-2 laboratories and facilitate high-throughput screening, LASV_VSV_pv, competent for a single round of viral entry and infection, was constructed using the VSV backbone, and the GPC gene was replaced with a luciferase gene. An HTS flowchart is depicted in Figure 1A. Inhibitors were defined as prime candidates when luciferase inhibition was >50% and there was no apparent cytotoxicity at a concentration of 100 μM (Figure 1B). The reconfirmed screening was then carried out over a broader concentration range, and four compounds were identified as candidates based on their dose-dependent inhibition and cell viability > 80% at the highest tested concentration (800 μM). Then, a counter screen was executed to exclude compounds exerting inhibition by targeting the VSV backbone; F1920 (1-Hydroxy-2,3-dimethylbenzene) and F1965 (1-(4-Chlorophenyl)-1-cyclopropanecarbonitrile) were selected as both compounds showed robust inhibition of LASV_VSV_pv, whereas they had little effect in VSVpv. The 50% inhibitory concentrations (IC_50_s) of F1920 and F1965 against LASV_VSV_pv on Vero cells were 63.60 and 44.13 μM, respectively, whereas the values obtained for A549 cells were 21.50 and 72.15 μM, respectively, indicating that both compounds exhibit similar inhibitory effects in different cell types (Figure 1C,D). The IC_50_ values are relatively high, which may be because the fragment drug molecules are very small (F1920 122 g/mol, F1965 177 g/mol), roughly 1/5 to 1/3 of that of a normal drug, suggesting the potential for continued growth and optimization of the backbone.

### 3.2. F1920 and F1965 Inhibit LASV GPC-Mediated Membrane Fusion

To examine the effects of both hits on the LASV GPC-mediated membrane fusion, a qualitative assay was performed. As shown in Figure 2A, low pH treatment of GPC transfected cells induced obvious membrane fusion and loss of cellular boundaries. Both F1920 and F1965 prevented the low-pH-triggered membrane fusion. Although the formation of small syncytia was still observed at all concentrations tested, cell boundaries were evident, and unfused single cells were the majority in the field of view.

To quantitatively evaluate the inhibitory activities, the fusion efficacy was determined using a dual-luciferase assay [16,17,18]. As shown in Figure 2B, both compounds inhibited GPC-mediated membrane fusion in a dose-dependent manner, suggesting that they inhibit GPC conformational changes induced by an acidic environment. Notably, the inhibition of membrane fusion by both hits was similar to their inhibitory effects on LASV_VSV_pv entry, suggesting that both compounds prominently inhibited GPC-mediated membrane fusion.

To further examine whether both compounds could bind to GPC and exert a virucidal effect, a virucidal assay was performed [19]. As shown in Figure 2C, luciferase activity was not suppressed in either group, indicating that both hits had little virucidal effect on LASV_VSV_pv. Then, the effects of the hits on binding were studied using the replication-competent LASV recombinant virus (LASV_VSV_rv), which was also constructed with the VSV backbone, and the VSV-G was substituted with LASV-GPC. As shown in Figure 2D,E, the monoclonal antibody (MAb) IIH6, which recognizes a functional glycan epitope on α-DG and competes with LASV for binding to cells [20,21] and was used as a positive control, blocked LASV_VSV_rv infection at a concentration of 200 μg/mL. Notably, no significant decrease in GFP was observed in either of the hit-treated groups, suggesting that neither hit interferes with the receptor-binding subunit GP1 [22].

### 3.3. Mutations Located in GP2 Transmembrane Domain (TM) Confer Resistance to Hits

To study the viral targets of the hits, adaptive mutants were selected by serially passaging LASV_VSV_rv in the presence of either hit. As shown in Figure 3A, the F1920-treated group exhibited a transient change of I445 to S445, which emerged at P5, completed the substitution at P12, and finally reversed to I445 at P20. Similarly, in the F1965-treated group, the change of L442 to F442 was recorded at P12 and was reversed at P20. Notably, the change of F446 to L446 was stable in both hit-treated adaptive mutants (P20) and the reverse mutation was not observed, which was in line with our previous reports that F446L was found in serial passages with casticin and compound 57 [11,12]. Intriguingly, all the changes of amino acid residues (L442F, I445S, F446L) were located on the resistant side of the α-helix of GP2 TM, suggesting that some key residues on this side played critical roles in the regulation of resistance to the fusion inhibitors (Figure 3B,C) [12].

To further test whether these changes of amino acid residues conferred resistance to the hits, LASV_VSV_pv containing each mutant was constructed and drug sensitivity/resistance was studied. As shown in Figure 4A, all the mutants conferred resistance to both hits, as the luciferase activity was higher in the mutant group than in the WT group; that is, the inhibition against the mutant was less effective when treated with the same concentration of the hit.

As both hits inhibited LASV_VSV_pv entry by blocking GPC-mediated membrane fusion, we further investigated whether the mutants conferred resistance to membrane fusion inhibition. As shown in Figure 4B, GPCWT was sensitive to both hits, as a decrease in syncytium formation was observed with increasing drug concentration. Conversely, both GPCL442F and GPCI445S pro, as membrane fusion was inhibited by neither hit. These results showed that all three residues might serve as potential targets of the hit drugs and may be involved in the stabilization of the prefusion conformation of GPC, thus conferring resistance to the hits.

### 3.4. F1920 and F1965 Showed Antiviral Activity against Other Mammarena Viruses

To examine whether the hits could inhibit the entry of other arenaviruses, pseudotypes of pathogenic viruses (including prototypes LCMV, LUJV, and the most closely related MOPV, JUNV, MACV, GTOV, SBAV, and CHAPV) and filoviruses (EBOV and MARV) were generated with the VSV backbone. As shown in Figure 5, F1920 effectively inhibited JUNV, GTOV, CHAPV, and LUJV pseudotype virus infections in a dose-dependent manner, whereas F1965 showed little to mild inhibition of the pathogenic arenavirus and filovirus pseudotype viruses, as the inhibition was <80%, even when treated with the highest tested concentration (400 μM), suggesting the potential to develop F1920 as a broad-spectrum antiviral drug.

### 3.5. Validation of the Antiviral Effects with Replication-Competent Recombinant Viruses

To validate the antiviral effects of the hits, replication-competent recombinant viruses with VSV and LCMV backbones were constructed. As shown in Figure 6A, both F1920 and F1965 showed robust inhibition of LASV_VSV_rv infection and the inhibitory effects were similar to those against LASV_VSV_pv (Figure 1C,D). We next examined the antiviral effects using LASV_LCMV_rv. Although both hits could inhibit LASV_LCMV_rv infection, the degree of inhibition was milder than that against LASV_VSV_rv (Figure 6B). For example, at the highest tested concentration (400 μM), F1920 exerted only ~80% inhibition of LASV_LCMV_rv, whereas it almost completely suppressed LASV_VSV_rv (98.8%). Similarly, 400 μM F1965 inhibited LASV_LCMV_rv entry by ~60% and that of LASV_VSV_rv by 95.9%. Notably, the MOI used in the antiviral tests of both LASV_VSV_rv and LASV_LCMV_rv were the same (0.05). The antiviral effect of the hits against the authentic prototype arenavirus, LCMV, was further studied. As shown in Figure 6C, neither hit inhibited the LCMV infection.

## 4. Discussion

In this study, we screened an FBDD library and found that F1920 and F1965 inhibit LASV_VSV_pv entry. The FBDD library contains compounds that are small and low in molecular weight (~200 Da). The binding affinity and efficacy of hits to the targets are usually weak because the fragments comprise only a part of the binding pocket. After merging the hits with multiple branches, hit-to-lead elaboration might afford a nanomolar inhibitor [24]. To exclude the possibility that the identified hits exhibited a pan-antiviral effect, as the small molecules might bind to either the viral or host cellular proteins, a number of other compounds (e.g., Carvacrol [MW, 150 g/mol] and thymol [MW, 150 g/mol], structurally similar to either F1920 or F1965, were tested, and none of them showed promising inhibition of LASV entry. Furthermore, since F1920 only inhibited a fraction of the tested viruses and had no effect on the rest, and F1965 was LASV-specific, it could be concluded that both hits have promising antiviral effects on LASV. The limitation of this work was the lack of data on authentic LASV. Notably, we also tested the drug–drug interaction between F1920 and F1965. Neither synergistic nor combination effect was found between the two hits.

Previously, we screened different kinds of drug libraries to identify inhibitors targeting arenavirus GPC, thus blocking virus entry [4,10,11,12]. The F446L mutant was screened out at high frequencies, which is in line with other reports showing that change at this position confers resistance to distinct entry inhibitors [25,26]. Notably, in addition to F446L, the mutants L442F and I445S appeared transiently in consecutive passages and eventually reverted to WT. The F446L mutant emerged after P16, became dominant after P20, and never reverted to the WT after P25. Interestingly, all three mutants, whether transient or constant, showed resistance to both F1920 and F1965. The change of F446 to L446 seemed to confer the most resistance to both drugs, particularly F1920, which might be the reason the other two changes were eventually lost or reverted back to WT. Most recently, the structure of a complete native LASV GPC containing the TM domain was solved [2]. Based on the reported structure (PDB:7PUY), it was observed that all the key residues (L442, I445, and F446) are located at the very end of the GP2 TM domain, and reside at the same side of the TM helix. Whether the key residues are directly accessible to the inhibitor or whether they collaborate with each other or with other residues, and thus contribute to stabilizing the prefusion conformation of GPC, needs further investigation. Notably, all three residues (L442, I445, and F446) were conserved between LASV and LCMV (Figure 3C). Unexpectedly, neither F1920 nor F1965 inhibited the entry of LCMV_VSV_pv. It is possible that subtle differences between LASV and LCMV GPC contribute to the differences in drug accessibility or that other residues are synergistically involved in drug sensitivity. In total, this study identified F1920 and F1965 as LASV entry inhibitors, highlighting the potential of these two fragment compounds for the development of lead broad-spectrum antiviral drugs. This study also highlights the role of key residues in the GP2 TM.

## Figures and Tables

**Figure 1 viruses-14-02649-f001:**
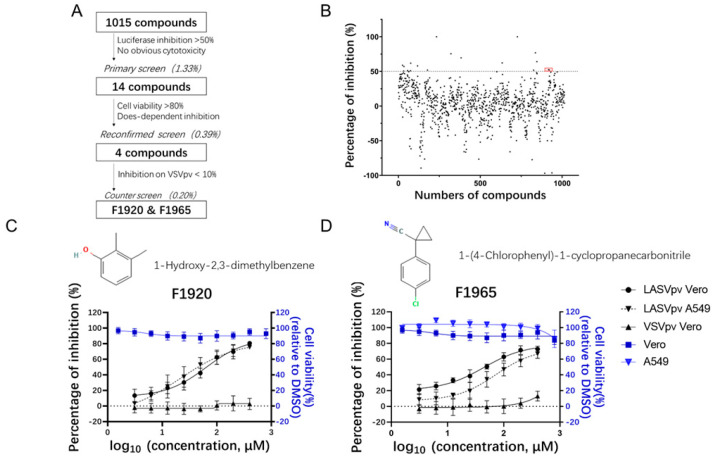
HTS for inhibitors of LASV entry from FBDD library. (**A**) HTS assay flowchart. (**B**) HTS of a library of 1015 fragment compounds for primary candidates inhibiting LASV_VSV_pv infection. Each dot represents the percent inhibition achieved with each compound at a concentration of 100 μM. F1920 and F1965 are highlighted in red. (**C**) Structure and dose–response curves of F1920 (1-Hydroxy-2,3-dimethylbenzene). (**D**) Structure and dose–response curves of F1965 (1-(4-Chlorophenyl)-1-cyclopropanecarbonitrile). Vero cells were treated in duplicate with F1920 or F1965 at the indicated concentrations, 1 h later, cells were infected with LASV_VSV_pv (MOI, 0.05) for 1 h. The infected cells were lysed 23 h later, and the luciferase activities were measured. Cell viability was evaluated using the CCK-8 assay. Data are presented as means ± SDs from 3–5 independent experiments.

**Figure 2 viruses-14-02649-f002:**
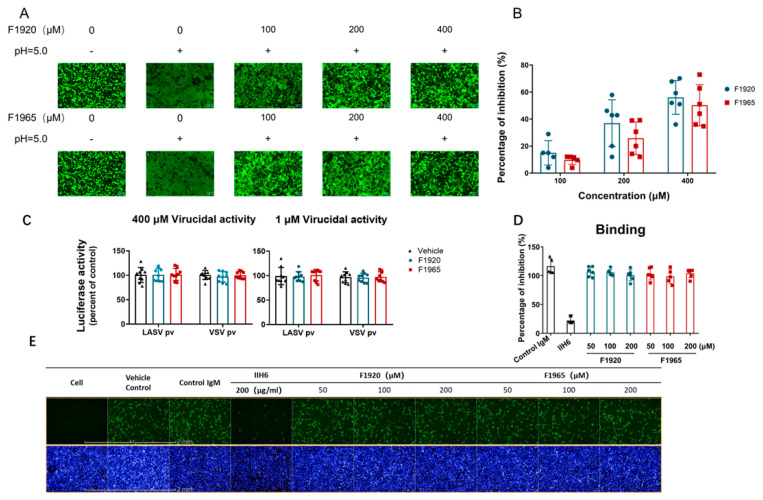
F1920 and F1965 inhibited LASV GPC-mediated membrane fusion. (**A**) Qualitative evaluation of the inhibitory activities of F1920 and F1965 on membrane fusion. 293T cells were co-transfected with pCAGGS-LASV GPC and eGFP; 24 h later, either compound was added for 2 h, followed by treatment with acidified DMEM (pH 5.0) for 15 min. The cells were then placed in a neutral pH DMEM. Syncytium formation was visualized 2 h later using fluorescent microscopy. Images are representative of 4 or 5 independent experiments. (**B**) A dual-luciferase assay was used to quantitatively evaluate the inhibitory activities of F1920 and F1965 on membrane fusion. 293T cells transfected with both pCAGGS-LASV GPC and pCAG-T7 were co-cultured at a ratio of 3:1 with the targeted cells transfected with pT7EMCVLuc together with the pRL-CMV control vector. Cell fusion activity was quantitatively determined by measuring firefly luciferase activity expressed by pT7EMCVLuc and standardized with Rluc activity expressed by pRL-CMV. (**C**) Virucidal assay. LASV_VSV_pv at an MOI of 20 was incubated with vehicle or compounds (400 μM for 1 h), diluted 400-fold, and added to cells. As a control, LASV_VSV_pv at an MOI of 0.05 was incubated with vehicle or compounds (1 μM for 1 h) and then added to cells. The luciferase activity was measured 24 h later. (**D**,**E**) Effects of the compounds on LASV_VSV_rv binding. A549 cells were pre-incubated with the compounds or vehicle at 37 °C for 1 h, followed by incubation with LASV_VSV_rv (MOI, 0.5) in the presence or absence of the compounds at 4 °C for 1 h. Pre-incubation with MAb IIH6 or an unrelated mouse IgM was conducted at 4 °C for 1 h. After extensive washing with cold PBS, GFP-positive cells were counted using an Operetta high-content imaging system 12 h later. Data are presented as the mean ± SDs of three independent experiments.

**Figure 3 viruses-14-02649-f003:**
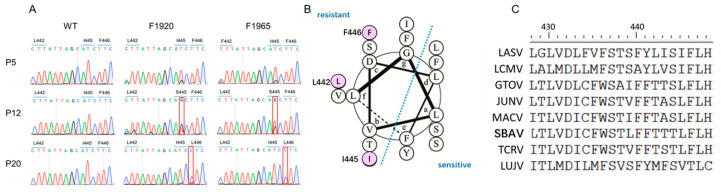
Selection of adaptive mutant. (**A**) Sequencing chromatograms of viruses from P5, P12, and P20. The adaptive changes of amino acid residues are highlighted in red box. (**B**) Helical-wheel project of LASV GP2 TM. The adaptive mutants were highlighted in purple. The project was drawn by using DrawCoil 1.0 [23]. (**C**) Amino acid sequence alignment of arenaviruses GP2 TM.

**Figure 4 viruses-14-02649-f004:**
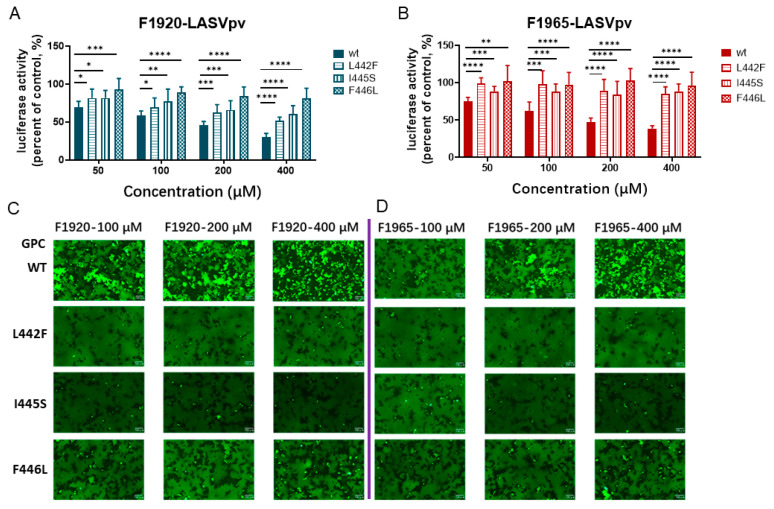
Resistant activities. (**A**) LASV_VSV_pv_L442F_, LASV_VSV_pv_I445S_, and LASV_VSV_pv_F446L_ conferred resistance to F1920. Vero cells were treated with F1920 at the indicated concentration; 1 h later, WT, LASV_VSV_pv_L442F_, LASV_VSV_pv_I445S_, and LASV_VSV_pv_F446L_ (MOI, 0.05) were added to the culture for 1 h. The luciferase activities were measured 23 h later. (**B**) LASV_VSV_pv_L442F_, LASV_VSV_pv_I445S_, and LASV_VSV_pv_F446L_ conferred resistance to F1965. Data are presented as means ± SD from three independent experiments. ****, *p* < 0.0001; ***, *p* < 0.001; **, *p* < 0.01; *, *p* < 0.05. (**C**) L442F, I445S, and F446L conferred membrane-resistance to F1920. 293T cells were transfected with pCAGGSGPCWT, pCAGGSGPCL442F, pCAGGSGPCI445S, and pCAGGSGPCF446L. After 24 h, F1920 was added for 2 h, followed by treatment with acidified (pH 5.0) DMEM for 15 min. Syncytium formation was visualized after 2 h. (**D**) L442F, I445S, and F446L conferred membrane-resistance to F1965. Images are representative fields from three independent experiments.

**Figure 5 viruses-14-02649-f005:**
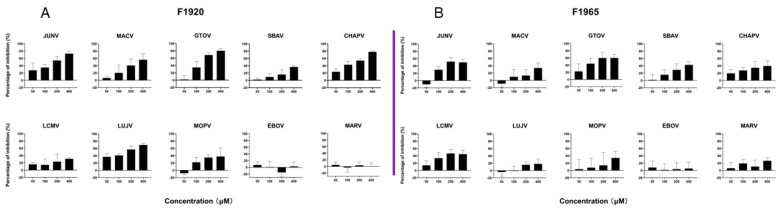
Broad-spectrum antiviral activity of both hits against pseudotype of the pathogenic mammarenaviruses and filoviruses. Vero cells were incubated in with either F1920 (**A**) or F1965 (**B**) for 1 h, then the pseudotypes of JUNV, MACV, GTOV, SBAV, CHAPV, LCMV, LUJV, MOPV, EBOV, and MARV were added at an MOI of 0.05. The supernatant was removed 1 h later, and the cell lysates were assessed for luciferase activity after 23 h. Data are presented as means ± SD from three independent experiments.

**Figure 6 viruses-14-02649-f006:**
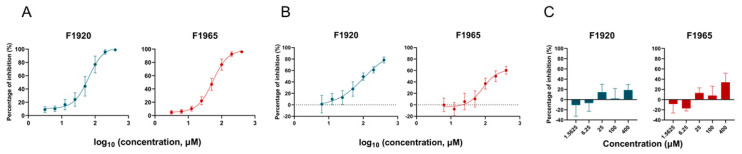
Inhibitory activity of F1920 and F1965 against the recombinant and authentic virus. (**A**) Inhibitory activity of F1920 and F1965 against LASV_VSV_rv based on VSV backbone. Vero cells were incubated with F1920 or F1965 for 1 h, and then LASV_VSV_rv was added at an MOI of 0.05 for additional 1 h. The supernatants were then replaced with either hit for 23 h and the GFP-positive cells were counted using an Operetta high-content imaging system. (**B**) Inhibitory activity of F1920 and F1965 against LASV_LCMV_rv based on LCMV backbone. Vero cells were incubated with F1920 or F1965 for 1 h, and then LASV_LCMV_rv was added at an MOI of 0.05 for additional 2 h. The supernatants were then replaced with either hit for 23 h and the GFP-positive cells were counted using an Operetta high-content imaging system. (**C**) Inhibitory activity of F1920 and F1965 against authentic LCMV. Vero cells were incubated F1920 or F1965 for 1 h, and then LCMV was added at an MOI of 0.01 for additional 1 h. The supernatants were then replaced with either hit for 23 h, and the cell lysates were subjected to RT-qPCR assay. Data are presented as means ± SD from 3–6 independent experiments.

## Data Availability

Not applicable.

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
