# Peer review of "Screening and Identification of Lassa Virus Entry Inhibitors from a Fragment-Based Drug Discovery Library"

_viruses, 2022, doi:10.3390/v14122649_

Round 1
Reviewer 1 Report
The manuscript by Hou et al. describes the identification of two entry inhibitors for Lassa virus as well as some other pathogenic arenaviruses. There is an urgent need for potential therapeutics for LASV and related viruses, and so the aim of the study is an important one. They were able to show that two fragment-based drugs from their library inhibited entry of pseudoviruses and recombinant viruses bearing LASV GPC. They were also able to show that specific amino acid changes could reverse the inhibition they saw by their drug candidates. The methodology used was sound in most instances, and they show that treatment reduces fusion via LASV GP2 and viral entry. The biggest limitation of the work for me is the lack of treatment using authentic LASV in vitro. I realize that this requires a BSL-4 facility, but this experiment, showing effective treatment and entry inhibition against LASV, is critical part of the study that was not done. I would hesitate to publish data on antiviral effects of drug candidates that does not include data with the authentic virus. With that data included, the study would provide a nice addition to the literature of possible drug candidates that could be tested in vivo or be used to study LASV GPC fusion and its inhibition.
I have a few other minor comments regarding the manuscript.
On line 204-205 is it possible that there is a higher IC50 for both drugs because their smaller size requires more drug present to interact with the GPC of the virus? I presume nothing is known about the molecular interactions of these compounds with the GPC aside from knowing what amino acids might be involved. I am wondering whether it may just be an issue of needing enough molecules to interact with GPC residues to prevent fusion.
In the discussion, it may be worth noting that it seems, looking at the data in figure 4, that the 446 AA change seems to confer the most resistance to both drugs, particularly F1920. This may be why the other two changes were eventually lost or reverted back to WT.
Minor comments/questions:
Line 86: when generating the recombinant VSVs, is there VSV glycoprotein incorporated into the particles? It says that the authors inoculated cells transfected with VSV G, so I am curious if the resulting recombinant particles incorporate VSV G? and if so, this may have an impact on the overall entry of the recombinant VSV compared with strictly VSV-LASV.
Line 109: should this be labelled as high throughout screening, rather than sequencing?
Line 126: should be co-transfected.
Line 148: here please put what the concentrations were of? Drug, Ab, etc.
For the PCR described in section 2.9, can the authors list the conditions that they used to run the RT-qPCR?
Line 232: it says that luciferase activity was suppressed here. Is this correct? In fig 2C, luciferase activity was the same in all treatments, so it was not suppressed?
line 266: Here and elsewhere the authors should not use mutation when referring to changes in amino acid residues. They should change these instances since mutations occur in the genome/nucleic acid sequence, resulting in AA changes.
In section 3.3, lines 275-287, I think there is some confusion regarding the labelling of figure panels for figure 4. The first section here should probably describe A and B? and the next, line 282, describe C? the figure legend also says there is a “D” panel, which is not in the figure.
Line 361: I would temper suggestions that treatments have antiviral effects against LASV until experiments with WT virus can be done. While these experiments suggest that these drugs can inhibit or reduce viral fusion, their antiviral activity against LASV remains to be seen.
Line 381: the same issue here, mentioning identification of LASV entry inhibitors. This was not done, and cannot necessarily be concluded from this study. Caveats should be mentioned.
Figure 1 should be formatted to include “F1920” over 1C similar to how 1D has “F1965” above it. So that it is clear each panel is for each drug.
Figure 2: I would strongly suggest suing scatter plots to show the data rather than bar graphs, so that readers can see all of the relevant data points for each treatment/infection. Bar graphs showing mean + SD can be misleading depending on the spread of the data.
Line 257: in the figure 2 legend, this should be the VSV LASV, correct, and no the LASVpv?
Reviewer 2 Report
I congratulate the authors for this very nice peace of work. While they found two fragments as good inhibitors of the viral entry, their IC50 values ate high uM range and thus I would like to see their synergistic or combination effect so that the dose requirement may be lowered or the antiviral effect gets increased.
Another suggestion is to try top few fragments together in variable combinations amd see the effect: it it looks promising then try to combine all those fragment to get a bigger but a better molecule containing all those shortlisted combination of fragments.
Mutational study is also impressive while it is hard to comment on them with current high IC50 values.
Round 2
Reviewer 1 Report
I want to thank the authors for their hard work in addressing all of the comments put forward. The revised manuscript is improved and presented more clearly. My original concern regarding the lack of authentic LASV still remains, and I still believe that the importance of the work is significantly reduced by not having that data. However given the logistical and practical issues with having those experiments done, I understand the want/need to publish these findings without them. The authors did add discussions of these caveats to the paper, so I think that suffices for now, until authentic virus assays can be completed.